# Mathematical model of COVID-19 intervention scenarios for São Paulo—Brazil

Osmar Pinto Neto [1,2,3 ✉], Deanna M. Kennedy[4], José Clark Reis[2], Yiyu Wang[4], Ana Carolina Brisola Brizzi [1,2], Gustavo José Zambrano[2], Joabe Marcos de Souza [2,5], Wellington Pedroso[1,2], Rodrigo Cunha de Mello Pedreiro[1,6,7], Bruno de Matos Brizzi[2], Ellysson Oliveira Abinader[8] & Renato Amaro Zângaro[1,3]

With COVID-19 surging across the world, understanding the effectiveness of intervention strategies on transmission dynamics is of primary global health importance. Here, we develop and analyze an epidemiological compartmental model using multi-objective genetic algorithm design optimization to compare scenarios related to strategy type, the extent of social distancing, time window, and personal protection levels on the transmission dynamics of COVID-19 in São Paulo, Brazil. The results indicate that the optimal strategy for São Paulo is to reduce social distancing over time with a stepping-down reduction in the magnitude of social distancing every 80-days. Our results also indicate that the ability to reduce social distancing depends on a 5–10% increase in the current percentage of people strictly following protective guidelines, highlighting the importance of protective behavior in controlling the pandemic. Our framework can be extended to model transmission dynamics for other countries, regions, states, cities, and organizations.

[1] Biomedical Engineering Department, Anhembi Morumbi University, São Paulo, SP 04546-001, Brazil. [2] Arena235 Research Lab, São José dos Campos, SP 12246-876, Brazil. [3] Center for Innovation, Technology and Education – CITE, Parque Tecnológico, São José dos Campos, SP 12247-016, Brazil. [4] Department of Health and Kinesiology, Texas A&M University, College Station, TX 77843, USA. [5] Universidade de São Paulo, Departamento de Engenharia Aeronáutica, São Paulo, SP 05508-900, Brazil. [6] Estácio de Sá University, Nova Fribugo, RJ 28611-135, Brazil. [7] Santo Antônio de Pádua College, Santo Antônio de Pádua, RJ 28470-000, Brazil. [8] Instituto Abinader, Manaus, AM 69057-015, Brazil. ✉email: osmarpintoneto@hotmail.com

 1

The World Health Organization (WHO) officially declared COVID-19 a pandemic in March of 2020 (ref. [1]). Eight months later, over 52 million people have tested positive for the virus resulting in more than 1.2 million deaths worldwide[2]. Most countries, including Brazil, have implemented widespread social distancing (SD) restrictions to mitigate the spread of COVID-19. It appears that these strategies can effectively reduce the number of cases and associated deaths[3]. However, an Imperial College London report indicated that Brazil's SD measures have not been as effective at reducing its effective reproduction number ($R_t$) as it has in other countries[4]. $R_t$ represents the average number of secondary cases that result from the introduction of a single infectious case in a susceptible population[5]. In May 2020, the estimated value for São Paulo, Brazil was 1.46 (ref. [4]). An $R_t$ greater than 1 indicates that Brazil has not controlled the COVID-19 pandemic. In fact, in May 2020, Brazil was the global hotspot with the highest infection and death counts in Latin America, with the state of São Paulo considered one of the main hotspot[4,6]. Given the high incidence of COVID-19 infections and deaths in Brazil in general and São Paulo specifically, understanding the effectiveness of intervention strategies on transmission dynamics is of major global health importance.

Several factors may lead to differences in intervention strategies on COVID-19 infection and death rates for São Paulo, Brazil, compared to other states and countries. For example, research has indicated the older adults (60+) are at the most significant risk of experiencing complications from COVID-19 (ref. [7]). Given that Brazil has a large percentage of its population over the age of 60, especially in urban areas such as São Paulo, aggravated cases may be exceptionally high compared to places with younger population demographics[8]. It has also been debated whether environmental conditions influence the behavior of the COVID-19 virus as the common cold and flu[9,10]. The majority of the research investigating the virus has been conducted in environments dissimilar to Brazil and other areas in the southern hemisphere. In addition, Brazil faces many economical and socio-cultural challenges that may affect mitigation strategies differently than cities, states, and countries that are the current focus of most prediction models[8]. Therefore, the purpose of the present investigation is to model different COVID-19 SD intervention strategies on transmission dynamics in São Paulo, Brazil, to determine the optimal strategy to contain the pandemic.

## Results

Our model has accurately fit the corrected 7-day running average of daily cases and daily deaths data for Brazil and São Paulo (Fig. 1). On the day of the analysis (11 May 2020), SD was estimated considering the reduction in people's overall mobility trends, at 41% and 52%, and the percentage ratio of the unsusceptible or protected people over the whole population (protection) was between 59–64% and 60–65%, for Brazil and São Paulo, respectively. Additionally, $R_t$ was 1.33 and 1.31, infection fatality ratio (IFR) was 0.46% and 0.53%, the attack rate was 2.2% and 3.5%, the latent period was 1.1 and 0.5 day, the infectious period was 14.4 and 9.2 days, hospitalization period was 4 and 4.1, and ICU period was 10 and 9.2 days, for the country and the state, respectively. All optimized coefficients estimated from the model can be found in Table 1.

Considering the country's size and that many regions face different stages of the pandemic, we concentrate further analyses on the state of São Paulo. However, note that our model can be applied to other countries, regions, states, cities, and organizations. The results suggest that the optimal strategy was the stepping-down strategy (Figs. 2a, b and 3a–h), and the optimal time window was 80 days (Figs. 2c and 3i–p). The results suggest that by constraining SD and protection levels to realistic ranges (30–70% and 50–70%, respectively), optimal solutions regarding strategies and time window to contain the first and second peak of the pandemic converge, confirming the optimal strategy (Figs. 2b and 4b, thick blue line) and window (Fig. 2c).

**The optimal solution for São Paulo.** Suppose current levels of SD (i.e. 41% for Brazil and 52% for São Paulo) and protection (i.e. 59–64% for Brazil and 60–65% for São Paulo) values are maintained. In that case, the results indicate that it is possible to contain the first peak in the pandemic using the optimal strategy (i.e., stepping-down) and time window (i.e., 80-day); however, the results also indicated that a second peak would be probable (Figs. 3j and 4a), unless current levels of SD are not maintained indefinitely (Fig. 4a, thin brown line). The optimal strategy and time window scenarios, in which SD was dropped or oscillated below the current levels, resulted in approximately 20,000 critical cases over the current ICU threshold. The results also indicated that an 80-day window stepping-down strategy could reduce the number of critical cases over the ICU threshold by approximately 90,000 compared to 80-day intermittent and constant strategies. Besides, the results indicated that a stepping-down strategy using an 80-day time window would reduce the number of critical cases over the ICU threshold by approximately 70,000 compared to a 40-day window and by approximately 30,000 compared to a 60-day window. If protection is maintained at current levels, the results indicate that it is possible to contain the first peak of the pandemic and significantly reduce the total number of cases through October 2020 if lockdown levels of SD (75%) were enforced for 60 days followed by a 40-day window stepping-down strategy. However, in a lockdown scenario, if the protection level drops below current levels with time, it may result in a sizeable second peak (Fig. 4a, yellow line).

Notably, the results indicate that it is possible to contain the first and second peaks of the pandemic in São Paulo, while maintaining current levels of SD (well below lockdown levels). For that, the current percentage of people (approx. 60%) strictly following protective guidelines has to increase by approximately 5–10%. The state should also adopt the optimal stepping-down strategy with a time window between 60 and 80 days (Fig. 3). After 60 days, a reduction in average SD across the remainder of the pandemic with a 60–80-day window stepping strategy would cause approximately a 30,000 decrease in the number of critical cases over the ICU threshold compared to the 80-day window intermittent strategy and 100,000 compared to the constant SD strategy. Additionally, considering the stepping strategy, a 60–80-day window would cause an approximately 40,000 decrease in the number of critical cases over the ICU threshold compared to a 40-day window.

If protection is increased by 10%, but SD is dropped from 52% to 40%, the optimal strategy and optimal time window suggest that critical cases over the ICU threshold would be approximately 30,000. The 80-day window stepping-down strategy would cause approximately an 11,000 decrease in the number of critical cases over the ICU threshold compared to the intermittent strategy and a 100,000 reduction compared to the constant SD strategy (Fig. 4c). Additionally, the stepping-down strategy with an 80-day window would cause an approximately 50,000 decrease in the number of critical cases over the ICU threshold compared to a 40-day window and a 15,000 decrease compared to a 60-day window.

Interestingly, the results suggest that protection has a larger influence on the pandemic's second peak than SD (Fig. 5). If protection dropped to 50% of its current value while maintaining the current quarantine SD level of 52% in São Paulo indefinitely, the pandemic would not be contained (Fig. 5a–h).

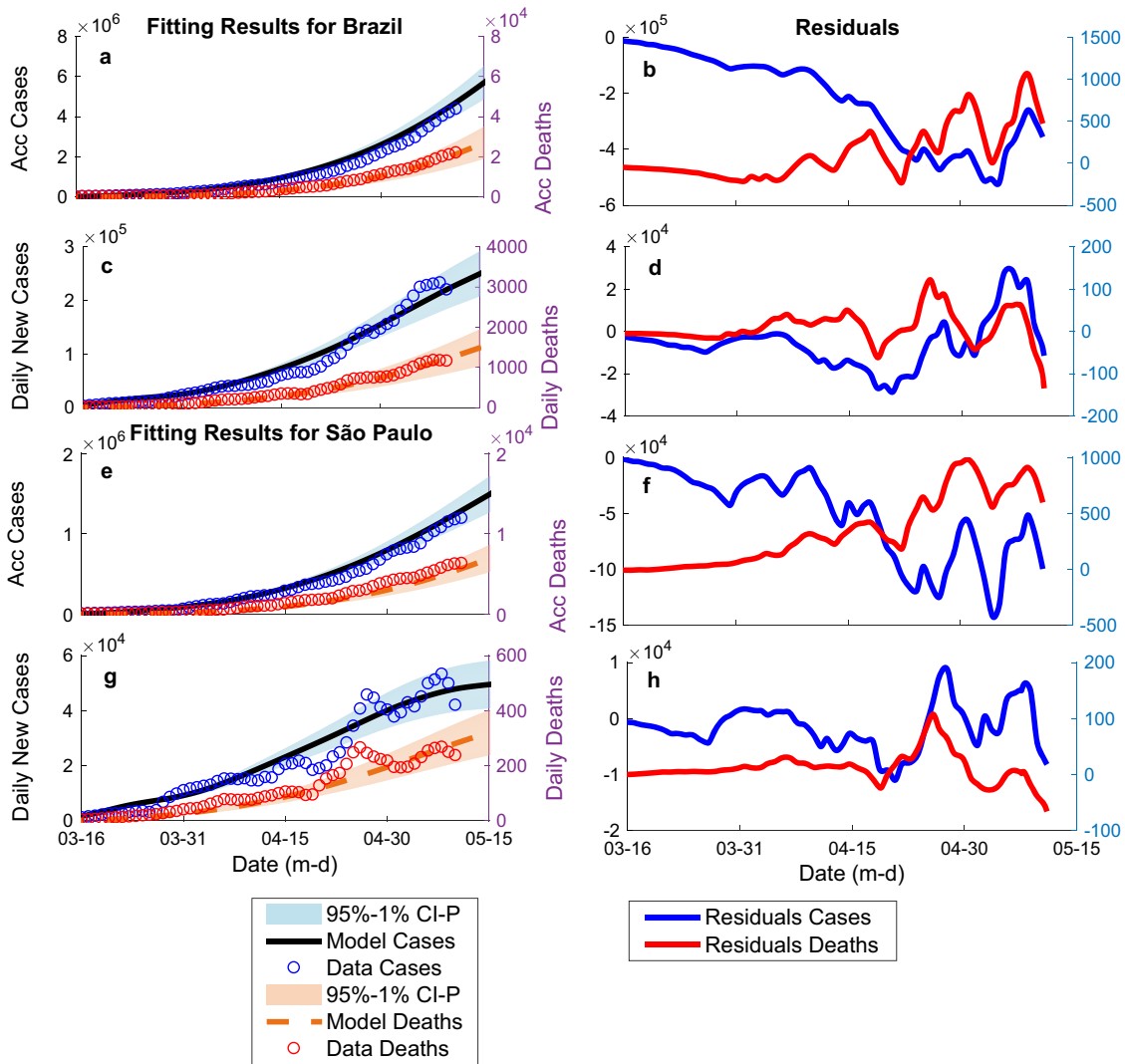

**Fig. 1 Fitting results for cases and deaths for Brazil and São Paulo. a** Corrected accumulated (Acc) cases and deaths for Brazil. **b** Acc data fitting residuals for Brazil. **c** Corrected daily new cases and deaths for Brazil. **d** Daily data fitting residuals for Brazil. **e** Corrected Acc cases and deaths for São Paulo. **f** Acc data fitting residuals for São Paulo. **g** Corrected daily new cases and deaths for São Paulo. **h** Daily data fitting residuals for São Paulo. Black lines represent the best-fit model expected cases; red dashed lines represent the best-fit model expected deaths. Blue circles are COVID-19 official data cases' data, and red circles are official deaths, both corrected by sub-testing factors. Blue and red shaded regions show confidence intervals of 95%, considering the 2.5 and 97.5% quantiles of the distribution of $n = 300$ uniformly distributed 1% errors or perturbations done to the best-fit model parameters.

**Table 1 Optimized coefficients for Brazil (BR) and the state of São Paulo (SP) on 11 May 2020, with respective ranges (lower boundary and upper boundary).**

| Coeffs | BR | SP | Lower B. | Upper B. |
|---|---|---|---|---|
| $\alpha$ | 0.024 | 0.018 | 0.01 | 0.12 |
| $\beta$ | 0.59 | 0.65 | 0.5 | 1.2 |
| $\gamma$ | 0.93 | 1.84 | 0.5 | 5 |
| $\delta$ | 0.07 | 0.11 | 0.07 | 0.5 |
| $\zeta$ | 0.25 | 0.24 | 0.2 | 0.33 |
| $\varepsilon$ | 0.10 | 0.11 | 0.05 | 0.14 |
| $m$ | 0.94 | 0.95 | 0.65 | 0.99 |
| $c$ | 0.36 | 0.35 | 0.25 | 0.5 |
| $f$ | 0.49 | 0.50 | 0.4 | 0.6 |
| $E0$ | 273 | 38 | $E0/2$ | $2E0$ |
| $I0$ | 195 | 18 | $I0/2$ | $2I0$ |
| $H0$ | 205 | 22 | $H0/2$ | $2H0$ |
| $C0$ | 0 | 0 | $C0/2$ | $2C0$ |
| $Re0$ | 0 | 0 | $Re0/2$ | $2Re0$ |
| $D0$ | 0 | 0 | $D0/2$ | $2D0$ |

## Discussion

We used a SUEIHCDR compartmental model to project thousands of scenarios for the transmission dynamics of COVID-19 in São Paulo, Brazil, through the next 2 years. We used genetic algorithms to optimize plans related to strategy type, SD magnitude, time window, and personal protection level. The goal was to determine the best-case scenario to control infections' current peak and avoid a second pandemic wave.

**Controlling the first peak.** We estimated Brazil still has an $R_t$ value over 1 (1.33 for the country and 1.31 for the state of São Paulo), indicating that it has yet to contain the first peak in infections and associated deaths caused by the COVID-19. Data from around the world (e.g., Asia, Europe, and North America) indicate that it is possible to mitigate the spread of COVID-19 with widespread SD and PPM measures[3]. However, our analysis of location data[11,12] suggests that the current level of SD is at only 42% for Brazil and 52% for São Paulo. In addition, the current

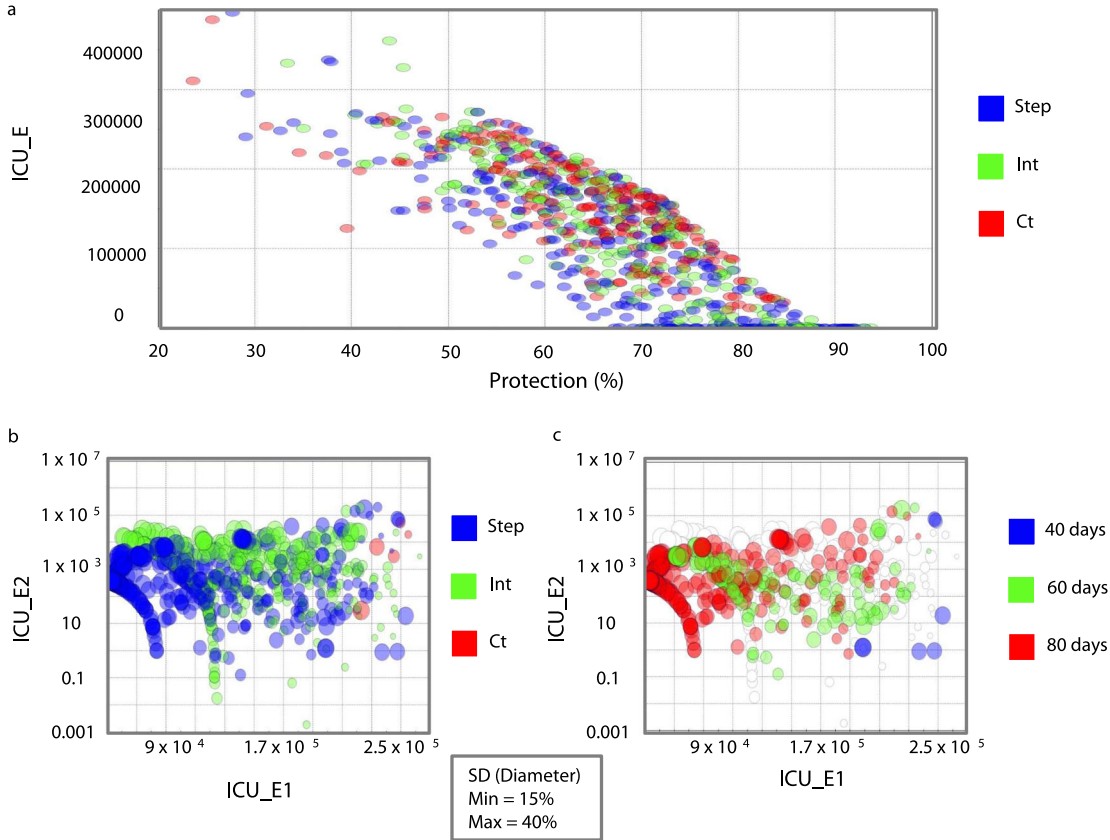

**Fig. 2 Mitigation strategies optimization results for the state of São Paulo. a** Design of experiment results showing the influence of social distancing (SD) strategy (red color indicates constant (Ct) SD strategy; green color indicates intermittent (Int) SD strategy; blue color indicates stepping down (Step) SD strategy) and the percentage ratio of the unsusceptible or protected people over the whole population (Protection), x-axis, on the total number of critical cases over the ICU threshold (ICU_E), y-axis, for the entire period of analysis (end-day 25 December 2021). **b** Constrained optimization results showing the influence of strategy (color) and SD magnitude (circle diameters corresponding to variations between 15 and 40%) on the number of critical cases over ICU threshold for the first (ICU_E1), x-axis, and the second peak of the pandemic (ICU_E2), y-axis. **c** Constrained optimization results showing the influence of Window size (red color indicates 80-day windows; green color indicates 60-day windows; blue color indicates 80-day windows) and SD magnitude (circle diameters) on ICU_E1, x-axis, and ICU_E2, y-axis.

protection values are 60–65% for the state and 59–64% for the country. Not only does our model indicate that current SD and protection values are insufficient in controlling the pandemic, but they will also have dire consequences on the overall number of infections and associated deaths with a large first peak. Our model also suggests that this will result in the need for public health resources, especially ICU, exceeding what is currently available. With the current levels of SD and protection, our model predicts that the ICU needed will surpass the available ICU in São Paulo at the first drop of SD levels, unless protection increases (Fig. 1). The scenario might be even worse because not all ICU beds available are exclusively dedicated to covid-19 patients and because some cities may experience the healthcare system failure before others. On the contrary, the state can invest in expanding ICU capacity, and with the continuous scientific development and increased medical experience, early treatment of the disease may improve; both factors would help minimize the number of patients surpassing the available ICU.

According to our model, it is possible for the state of São Paulo to gain control over the first peak, even without additional ICU beds or improvements in treatment, if the percentage of people strictly following protective guidelines significantly increased (at least 5–10%). Alternatively, immediately increase SD to values over 75% may solve the first peak problem. However, increasing

SD to values in 75% may result in a second peak if the restrictions are lifted within 40 days. SD at 75% represents a complete lockdown[13]. Thus, such a strategy should be used with caution. Further, it should only be used in critical cities and not the state as a whole. Our model suggests that if this high level of SD is used, it is only necessary for a short period.

Note that without a vaccine, however, low levels of SD may need to be in place for years to come to maintain control over the pandemic[14,15]. Widespread use of PPM (e.g., wearing facemasks, frequently washing hands, using hand sanitizer, maintaining a physical distance between other people, and avoiding agglomerations) and high testing rates have been emphasized to mitigating the spread of COVID-19 in addition to SD[15–19]. The results of our model agree. Our model indicates that increasing the percentage of people strictly following protective guidelines (e.g., wearing facemasks, proper hand hygiene, and avoiding agglomeration) to a level of 65–70% for São Paulo in combination with SD is necessary to contain the first peak in infections and associated deaths. Eikenberry et al.[20] estimated that the efficacy of using facemasks ranges between 50 and 90%, depending upon mask material and fitting[20]. They assumed that at least 50% protection factor would be achievable for well-made and well-fitted mask usage by the entire population[20]. The fitting of our model predicts that the current level of protection is about 60%.

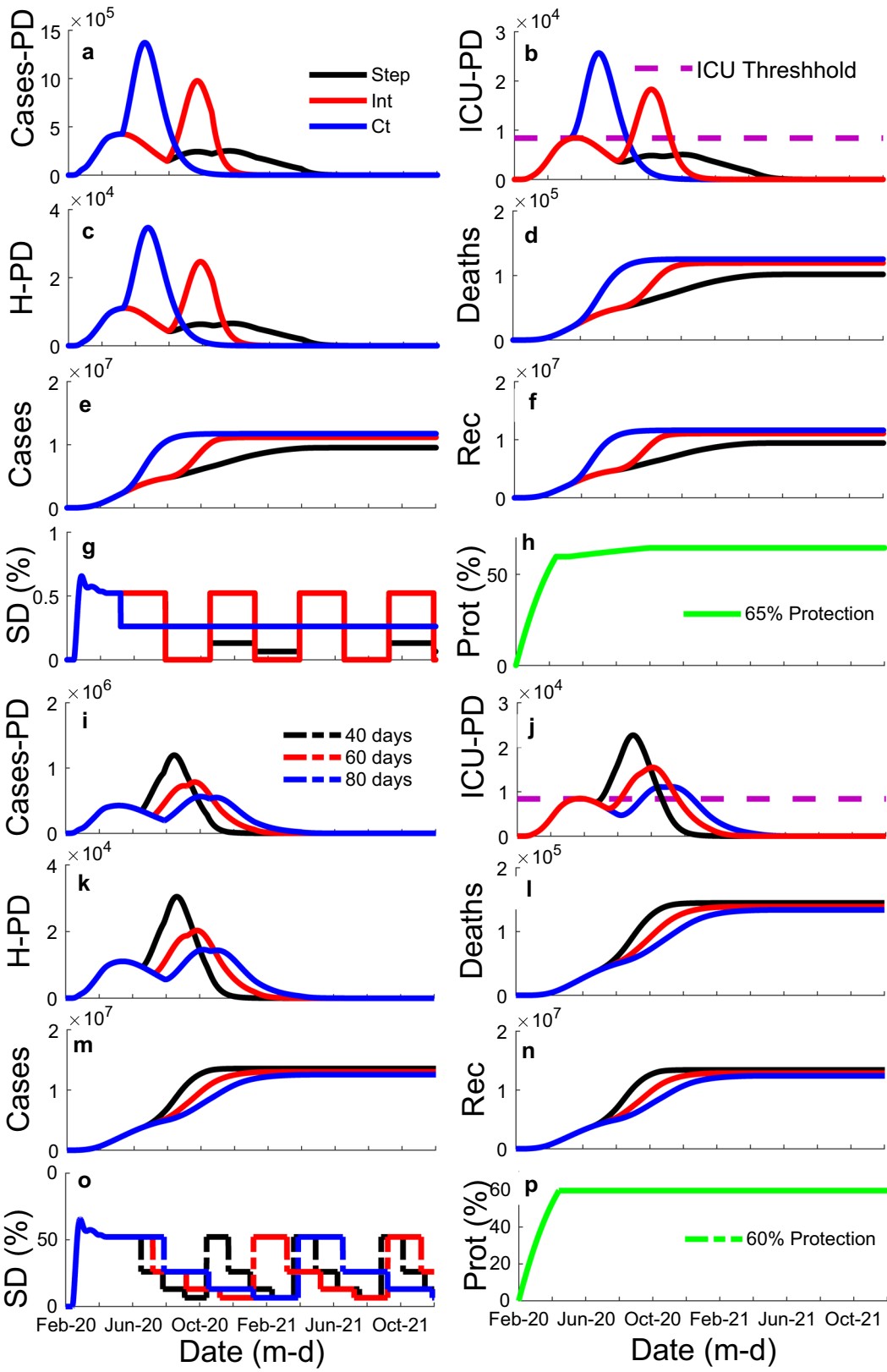

An increase in protection levels would require a massive effort by public health officials to enforce or educate people to use a facemask and maintain a 2 m safe distance from other people. Note, some Brazilian cities have implemented strict SD guidelines and require some level of personal protection[19]. For example, Belo Horizonte, Rio de Janeiro, and Salvador require facemasks in

public[21]. São Paulo recently announced they would require facemask as well.

**Avoiding a second peak.** If Brazil can effectively reduce $R_t$ below 1 with SD and personal protection, it is crucial to take steps to

**Fig. 3 Illustration of the influence of social distancing (SD) variation strategies (stepping down, intermittent, and constant) and different time windows (40, 60, and 80 days) on the model results for the state of São Paulo.** It shows influence of strategy (colored lines: black—stepping down (step); red—intermittent (Int); blue—constant (Ct)) through time, x-axis, on: **a** the number of cases per day (Cases-PD); **b** the number of estimated intensive care unit patients per day (ICU-PD); **c** the number of estimated hospitalized patients per day (H-PD); **d** the number of estimated accumulated deaths; **e** the number of accumulated cases; and **f** the number of recuperated cases. **g** Manipulations of SD through time, illustrating the three different strategies tested. **h** The percentage ratio of the unsusceptible or protected people over the whole population (protection—Prot (%)) through time for graphs **a**–**f**. It also shows the influence of different time windows (colored lines: black—40-days; red—60 days; blue—80 days)) through time, x-axis, on: **i** Cases-PD; **j** ICU-PD; **k** H-PD; **l** accumulated deaths; **m** accumulated cases; and **n** recuperated cases. **o** Manipulations of SD through time for graphs **i**–**n**, all three curves represent a stepping down strategy, but with three different time window sizes. **p** Protection through time for graphs **i**–**n**.

reduce the likelihood of a second peak. Our models indicate that if SD and personal protection measures are stopped too soon or decreased too much after the first peak containment, the second peak in infections and associated deaths will occur. Many experts agree that a secondary pandemic wave is likely if SD restrictions are lifted too quickly[22,23]. Therefore, determining when and how to relax restrictions has become the focus of epidemiological work worldwide. It has been proposed that a responsible exit strategy should continue SD restrictions alongside widespread testing and contract tracing[24]. Unfortunately, both these guidelines are hard to achieve in Brazil considering the economic pressure for re-opening[24] and that Brazil's testing efforts have been deficient. Brazil has conducted the least COVID-19 tests per capita worldwide[25]. In addition, reports indicate that Brazil uses sub-standard testing kits and only tracks hospitalized cases[26]. The lack of data for contact tracing in Brazil suggests efforts may be insufficient; further studies on the efficiency of such measures should be made. Note, however, if Brazil sufficiently reduces the number of infections to a more manageable level, testing and contact tracing may substantially improve. Importantly, our results suggest that the most effective way to reduce the number of people infected is protection (Fig. 5). Thus, we strongly urge the Brazilian government to mandate proper protection for its citizens.

Furthermore, our analysis suggests that Brazil can avoid a second peak, considering the results for São Paulo. According to our results, the best-case exit strategy to prevent the second peak in São Paulo was a stepping-down strategy over 2 years. A stepping-down approach would involve a gradual stair-step down. For example, the stepping-down approach we modeled multiplied SD values of 40, 30, and 20% by 1, then 1/2, then 1/4, then 1/8, and then back to 1, and then we repeated each stair-step down. A stepping-down strategy was also the best-case exit strategy modeled for the US[15]. A stepping-down approach may be beneficial because it allows for transmission periods, leading to herd immunity without overwhelming public health resources[14,27]. Alternatively, a one-time SD may result in a catastrophic second peak (Fig. 4a, yellow curve), if the virus reoccurred and not enough people have immunity[14,27]. Besides, the stepping-down approach resulted in a 6.5% reduction in total time required to SD over the 2 years, potentially reducing the economic and social costs associated with SD.

The current investigation results suggest that an 80-day window between each step was the most beneficial strategy. This result is also consistent with that for the US[15]. However, the current investigation modeled more time windows and more complex algorithms than the US model. The best-case stepping-down strategy, with an 80-day window, was a magnitude of SD of at least 50%, in the highest windows, or approximately 24% average through the period of analysis. Furthermore, the results suggest that higher protective measures could account for lower SD values, which may be associated with economic and psychological benefits. Alternatively, higher SD values delay the onset of the second peak. Delaying the onset of the peak may

allow Brazil to procure additional public resources and more time for the development of effective treatments or even a vaccine. The minimum protection percentage for this scenario was 65–70% for São Paulo. If protection is maintained at these levels, the models suggest that a second peak can be avoided. However, if protection drops, a second peak that crosses the threshold for available public resources could occur, perhaps with devastating consequences. Mortality rates associated with COVID-19 may rise when hospitals become overwhelmed and have fewer resources to treat patients with life-threatening symptoms[28]. This result supports the notion that personal protection is critical for maintaining control over the COVID-19 pandemic[15–19].

Overall, our results suggest that there are SD strategies other than lockdown and intermittent strategies in gaining and maintain control over the COVID-19 pandemic. Our results indicate that a stepping-down strategy may be more effective in controlling the pandemic than the more suggested constant and intermittent strategies. Our results also highlight the importance of protection in controlling the pandemic. Given that high levels of SD are associated with many economic and psychological consequences, strategies that can reduce the magnitude of SD over time while maintaining control of the pandemic is of great importance to global health. Our results suggest that it is possible to reduce SD over time if people engage in high levels of protection such as wearing facemasks, using proper hand hygiene, and avoiding agglomerations. Such behavior may be the most effective strategy in combating the COVID-19 pandemic, especially in countries with limited public health resources. While our analysis was focused on São Paulo, it is essential to note that our methods can model other countries, regions, states, cities, and organizations. Besides, the results of the current investigation are of global importance and may have universal application. As such, we suggest that our framework and strategies be tested in other countries and states. Note the globally available mobility data were used to calculate SD.

## Methods

**The SUEIHCDR model.** We extended a generalized Susceptible-Exposed-Infected-Recovered (SEIR) compartmental model and analysis[15] using factors specific to COVID-19 (refs. [29,30]) to investigate the COVID-19 pandemic in São Paulo, Brazil. Due to Brazil's continental size and the intrinsic properties of its different regions, we concentrate our analysis on the state of São Paulo. São Paulo is the most populous state in Brazil, and it is considered a significant hotspot within Brazil[6]. Our model assumes that the state of São Paulo is a single, homogeneously mixing system[31]. To verify this assumption, first, we confirmed that the COVID-19 daily cases and daily deaths in the state as a whole and the city of São Paulo were correlated (Supplementary Fig. 1). Second, we established that there is limited spatial variation in the state's pandemic progression by comparing the COVID-19 cases' across the state on 30 April, 15 May, and 30 May 2020 (Supplementary Fig. 2). Finally, we verified the state's homogeneity in terms of population density and that the state was isolated from other countries and Brazilian states during this pandemic period (Supplementary Figs. 3 and 4). Limitations of the homogeneously mixing system assumption are discussed next under the "Model limitations" subsection.

Our model comprises eight compartments: Susceptible, Unsusceptible, Exposed, Infected, Hospitalized, Critical, Dead, and Recovered (SUEIHCDR).

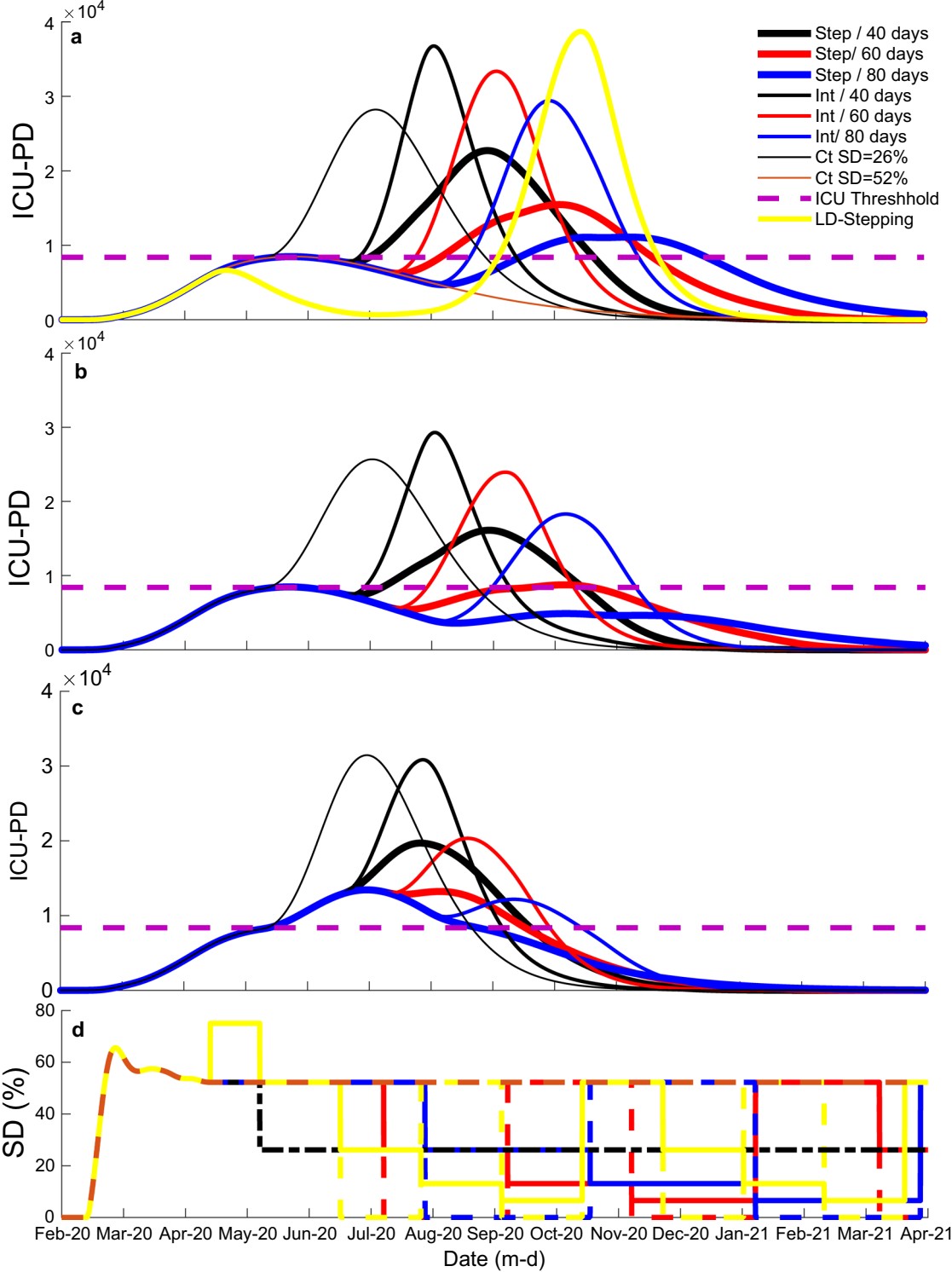

**Fig. 4 Representative model results for intensity care units bed occupancy per day (ICU_PD or critical cases) results for the state of São Paulo. a** Nine different scenarios showing the influence of strategy (line thickness: stepping down (step)—thick; intermittent (Int)—intermediate; constant (Ct)—thin; line color yellow: lockdown followed by stepping down—LD-stepping) and window size (line color: black—40-days; red—60 days; blue—80 days) with the percentage ratio of the unsusceptible people over the whole population (protection) kept at the current state-level of 60%; for the stepping and intermittent scenarios, social distancing (SD) maximum magnitude is kept at current levels, as estimated by mobility data (SD = 52%); for the LD-stepping scenario there is a 40-day 75% SD (lockdown) period followed by a 80-day stepping strategy with SD maximum magnitude of 52%. **b** Similar scenarios but with 70% protection, instead of 60%. **c** Similar scenarios with 70% protection and 40% maximum SD, instead of 52%. **d** Manipulations of SD through time, illustrating all scenarios tested. Dashed purple line indicates the total number of available ICU for the state.

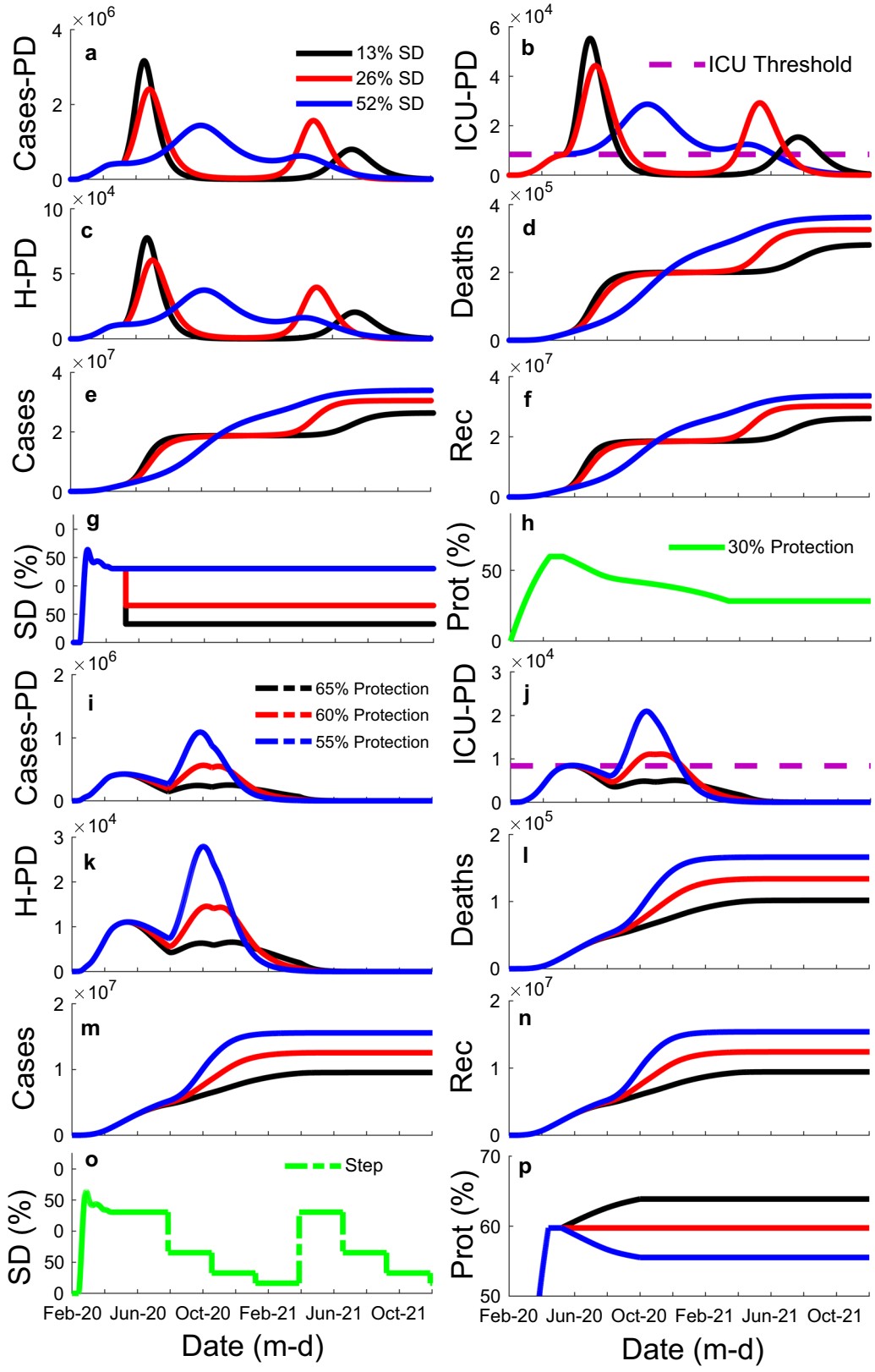

The SUEIHCDR model assumes that as time progresses, a susceptible person out of the population (Npop) (Eq. 1) can either become unsusceptible (Eq. 2) considering a protection rate ($\alpha$; Eq. 9) or exposed (Eq. 3) to the virus considering SD and an infection rate ($\beta$). This protection rate was introduced to account for possible decreases in the number of susceptible people to the virus caused by factors other than SD, such as the use of facemasks and hand washing.

Exposed people become infectious after an incubation time of $1/\gamma$ (Eq. 4). Infected people stay infected for $1/\delta$ and can recover with no medical attention ($m$) or can be hospitalized ($1-m$). Hospitalized people (Eq. 5) stay for $1/\zeta$ days and can either recover ($1-c$) or become critical ($c$) needing to go to an intensive care unit (ICU). A person stays on average $1/\varepsilon$ in the ICU (Eq. 6) and can either go back to the hospital ($1-f$) or die ($f$; Eq. 7). Recovered people (Eq. 8) can come

**Fig. 5 Illustration of the influence of different mean social distance (SD) magnitude (13, 26, and 52%) and different percentage ratios of the unsusceptible or protected people over the whole population (protection—65, 60, and 55%) on the model results for the state of São Paulo.** It shows influence of SD (colored lines: black—13%; red—26%; blue—52%) through time, x-axis, on: **a** the number of cases per day (Cases-PD); **b** the number of estimated Intensive Care Unit patients per day (ICU-PD); **c** the number of estimated Hospitalized patients per day (H-PD)); **d** the number of estimated accumulated deaths; **e** the number of accumulated cases; and **f** the number of recuperated cases. **g** Constant strategy SD curves illustrating the three different magnitude tested. **h** The percentage ratio of the unsusceptible or protected people over the whole population (protection—Prot (%)) through time for graphs **a-f**. It also shows the influence of different protection levels (colored lines: black—65% endpoint; red—60% endpoint; blue—55% endpoint) through time, x-axis, on: **i** Cases-PD; **j** ICU-PD; **k** H-PD; **l** accumulated deaths; **m** accumulated cases; and **n** recuperated cases. **o** Manipulation of SD through time for graphs **i–n**, a stepping down strategy. **p** The three protection levels tested in graphs **i–n**.

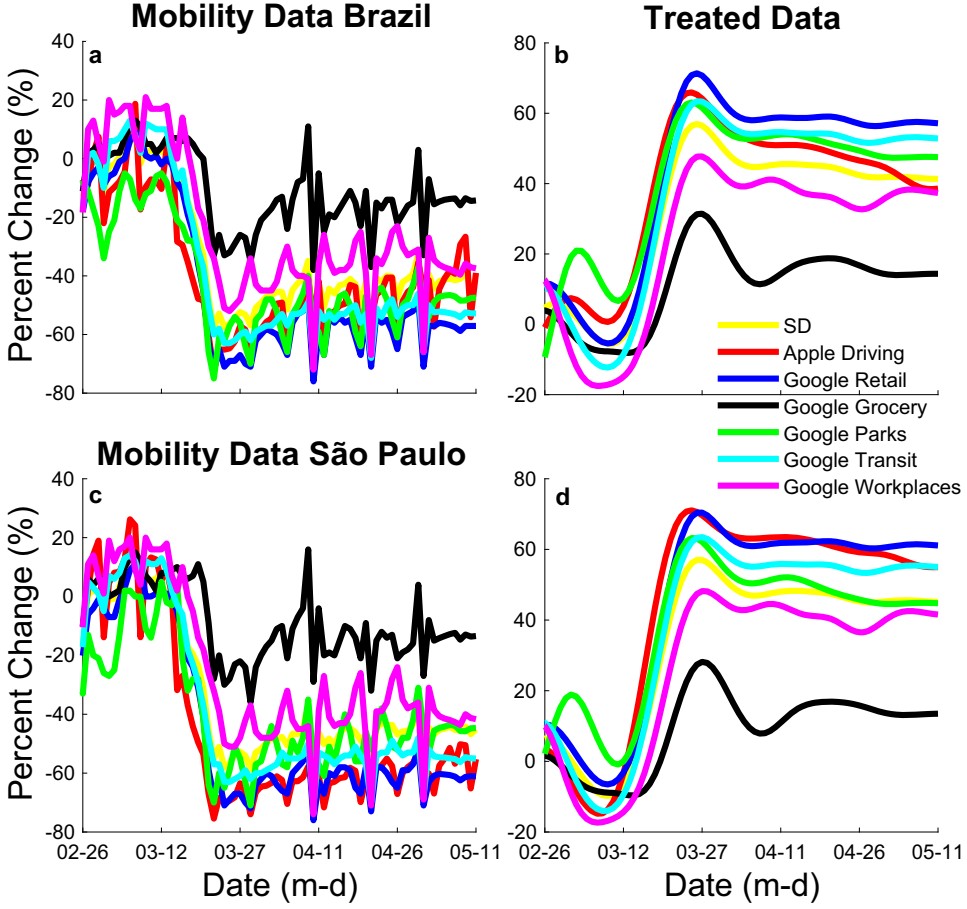

**Fig. 6 Mobility trends raw and treated data from Apple Maps and community mobility reports from Google. a** Mobility data for Brazil. **b** Treated mobility data for Brazil. **c** Mobility data for the state of São Paulo. **d** Treated mobility data for for the state of São Paulo. Data were low-pass filter filtered at 0.09 Hz (Butterworth fourth order), and percentage changes from baseline were considered (Apple's driving data were averaged with Google's retail and recreation, grocery and pharmacy, parks, transit stations, and workplaces). Social distancing (SD, yellow) was determined as the mean of Apple Driving (red) and Googles' Retail (blue), Grocery (Black), Parks (Green), Transit (Blue), and Workplaces (pink) treated data.

from infection ($m$) or the hospital ($1-c$). At last, the effective reproduction number $R_t$ (Eq. 10) of our model can be estimated as

$$\frac{dS(t)}{dt} = -\frac{(1-\text{SD}(t))\beta S(t)I(t)}{N_{\text{pop}}} - \alpha(t)S(t) \qquad (1)$$

$$\frac{dU(t)}{dt} = \alpha(t)S(t) \qquad (2)$$

$$\frac{dE(t)}{dt} = \frac{(1-\text{SD}(t))\beta S(t)I(t)}{N_{\text{pop}}} - \gamma E(t) \qquad (3)$$

$$\frac{dI(t)}{dt} = \gamma E(t) - \delta I(t) \qquad (4)$$

$$\frac{dH(t)}{dt} = (1-m)\delta I(t) + (1-f)\varepsilon C(t) - \zeta H(t) \qquad (5)$$

$$\frac{dC(t)}{dt} = c\zeta H(t) - \varepsilon C(t) \qquad (6)$$

$$\frac{dD(t)}{dt} = f\varepsilon C(t) \qquad (7)$$

$$\frac{dR(t)}{dt} = m\delta I(t) + (1-c)\zeta H(t) \qquad (8)$$

$$\alpha(t) = \alpha_0 \frac{\log(t+1)}{\log(t_f)} \qquad (9)$$

$$R_t(t) = \frac{(1-\text{SD}(t))\beta}{\delta}\left(1 - \frac{\int_0^t \alpha(t)S(t)}{N_{\text{pop}}}\right) \qquad (10)$$

where $\alpha_0$ is the maximum or minimum possible value for a window of time and $t_f$ is

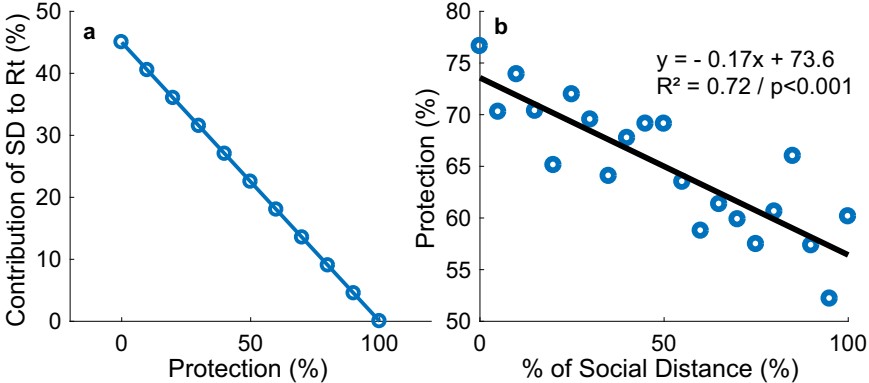

**Fig. 7 The relationship between social distancing (SD) and the percentage ratio of the unsusceptible or protected people over the whole population (Protection). a** It shows how the effect of a 45% SD in reducing effective reproduction number ($R_t$) as Protection percentage increases from 0 to 100%. **b** Sensitivity analysis determining the effect of changes in social distancing (SD) on protection percentage, considering 5% decrements to the magnitude of SD with respective linear regression equation, $y = -0.17x + 73.6$, and Pearson's correlation coefficient squared, $R^2 = 0.72$, $F_{1,19} = 48.891$, $p < 0.001$, 95% CI [−0.223, −0.120].

the window's final time. Alpha was optimized considering the window of time from the beginning of the pandemic until the present day. It was manipulated afterward in different windows of time to project possible future scenarios. Changes in $\alpha$ cause changes in the number of unsusceptible people to the disease at a given time; we exhibit $\alpha$ manipulations as the percentage ratio of the unsusceptible or protected people over the country's population (i.e. Protection (%)). Note that our model's insusceptibility accounts for the time-dependent state of an individual or behavior that can take someone from susceptibility to insusceptibility or the other way around.

Furthermore, SD until the present day was determined from mobility trends data from Apple Maps[11] and community mobility reports from Google[12] (Fig. 6a, c). Data were low-pass filtered at 0.09 Hz (Butterworth fourth order), and percentage changes from baseline were considered (Apple's driving data were averaged with Google's retail and recreation, grocery and pharmacy, parks, transit stations, and workplaces average percent change from baseline—Fig. 6b, d). From present-day forward, SD was manipulated in different windows of time to project possible future scenarios.

**Solving and testing the model**. We used the fourth-order Runge–Kutta numerical method to solve our system of ordinary differential equations in MATLAB (MathWorks Inc.R14a).

**Fitting of model's coefficients**. We used daily cases and daily deaths time series[2,32] with a 7-day running average to fit the model (Fig. 1). To account for sub-testing, both cases and deaths were corrected by a factor: 25.9 and 25.9 for cases, and 1.9 and 1.7 for deaths, for Brazil and São Paulo, respectively. Death sub-test factor was determined comparing data from deaths in the last months to average deaths in the same period from past 5 years[33–35]. The resulting sub-factor was similar to that reported by other sources[36]. Unlike the deaths' sub-test factor, there was no local data available to provide an accurate estimation of the cases' factor. Cases' sub-test factor was calculated as the ratio between the death rate in Iceland (the country with the greatest percentage of test per inhabitant[37]), corrected by age stratification (older adults (60+): 6.4%; senior older adults (80+): 13.4%[38]). Given the low testing rates in many countries, data from Iceland are believed to be a more accurate representation of global cases than other available sources[39]. Regarding age, our models' IFR values represent an overall average.

Fitting analysis was done with a custom build MATLAB global optimization algorithm using Monte Carlo iterations and multiple local minima searches (Fig. 1). The algorithm was tested for the best solution considering all inputs varying within ranges obtained from the WHO[40] and several publications[41–43] (Table 1). Infected initial values ($I0$) were determined from corrected accumulated cases and initial death values ($D0$). Other initial values were set proportional to $I0$ considering model parameters ($m$, $c$, $f$); all initial parameters could vary during optimization (Table 1). Data's start date was 25 February 2020 and the end date was 25 December 2021. Data under 50 active cases were discarded. Model results were based on an average of 5000 runs. Confidence intervals of 95% estimated considering the 2.5% and 97.5% quantiles of the distribution of $n = 300$ uniformly distributed 1% errors or perturbations to the model parameters to evaluate the confidence in the model results, to infer if future projected scenarios were statistically different at a 5% level, and to compare model results to actual ICU numbers. Note that confidence interval shaded areas were not included in all figures for clarity. Figures were plotted in MATLAB. Linear regression analysis was done in IBM SPSS Statistics for Windows (Version 20, IBM Corp). Peak ICU

estimations were compared to data obtained from the Brazilian government health database[44].

**Future projections**. After the model's coefficients were fitted, an optimization workflow using ESTECO's mode Frontier (Esteco s.p.a; 2017R4-5.6.0.1) was implemented to find the optimal mitigation strategy out of the proposed here could be identified. Future scenarios considered changes in SD and protection starting 1 June 2020. First, we run a DOE (Design of Experiment) scheme of around 1000 SOBOL individuals and 1000 Latin Hypercube individuals created by varying maximum SD values per window within 0–75% and protection percentage between 20 and 95%. Then data were constrained considering more realistic ranges of SD and protection (15–40% and 50–70%, respectively) (Supplementary Figs. 14 and 15). We propose three strategies of mitigation (Fig. 3g): (1) a stepping-down strategy (starting at a specific SD, it is divided by half for the next three time windows, on the fourth time window SD is back to its initial value, and the process is repeated); (2) a standard intermittent SD strategy (a specific SD value alternates with periods of no SD); and (3) a constant SD strategy (SD is kept constant at a specific value (Fig. 3g), with each strategy considering three different times windows: 40, 60, or 80 days (Fig. 3o). Furthermore, strategies were compared across different protection percentages (Fig. 5p) using similar average SD across time; note, however, that because of its design, when intermittent and stepping-down strategies have the same maximum SD, and when adopting half of these values for the constant SD strategy, average SD values across time tends to be 6.25% smaller for the stepping-down strategy compared to the other two. A MOGA—Multi-Objective Genetic Algorithm—was used to drive the optimization due to the discrete nature of the variables Strategy and Window. We minimize the number of critical cases over ICU threshold (ICU_E) for the duration of the analysis, the number of critical cases over ICU threshold in the first peak of the pandemic (ICUE_1), the number of critical cases over ICU threshold in the second peak of the pandemic (ICUE_2), and SD.

**Model limitations**. One limitation of our model is to assume mobility as a direct proxy for contact rates. Although it has been suggested there is a high correlation between the two[45], this relationship is complex. As can be seen from Eq. (10), the reproduction number is affected by SD and $\alpha$. From the equation, we can see that if $\alpha$ is zero (i.e. the population is doing no protective measures), the effect of SD in reducing the disease's transmission corresponds to its value. Nevertheless, our model assumes that at the beginning of the pandemic, when people are still not engaged in protective behavior, an increase in SD greatly impacts transmission. Such correspondence between SD as measured from mobility data from cellphones has been recently observed[46,47]. However, as time advances and $\alpha$ increases, the corresponding effect of SD in $R_t$ decreases. In the unlikely boundary case where protection reaches 100% (i.e. every person is doing every possible protective measure), SD does not affect $R_t$ as $R_t$ equals zero. Such a decrease in correlation between SD and $R_t$ has also been observed as some countries have started opening their economies. Although SD has linearly decreased[12], there was no corresponding increase in $R_t$, as seen in the daily new cases' curves (e.g. United Kingdom and Italy[2]). To illustrate, Fig. 7a shows how the effect of a 45% SD in reducing $R_t$ changes as protection increase from 0 to 100%. In practical terms, the first change in mobility occurred at the beginning of the pandemic, when people were not yet engaged in protective behavior. However, at the time of re-opening, people increased mobility, but with much greater caution and knowledge on reducing the possibility of infection. Given the possible inherent error of determining SD by noisy mobility data (Fig. 6), we ran a sensitivity analysis to

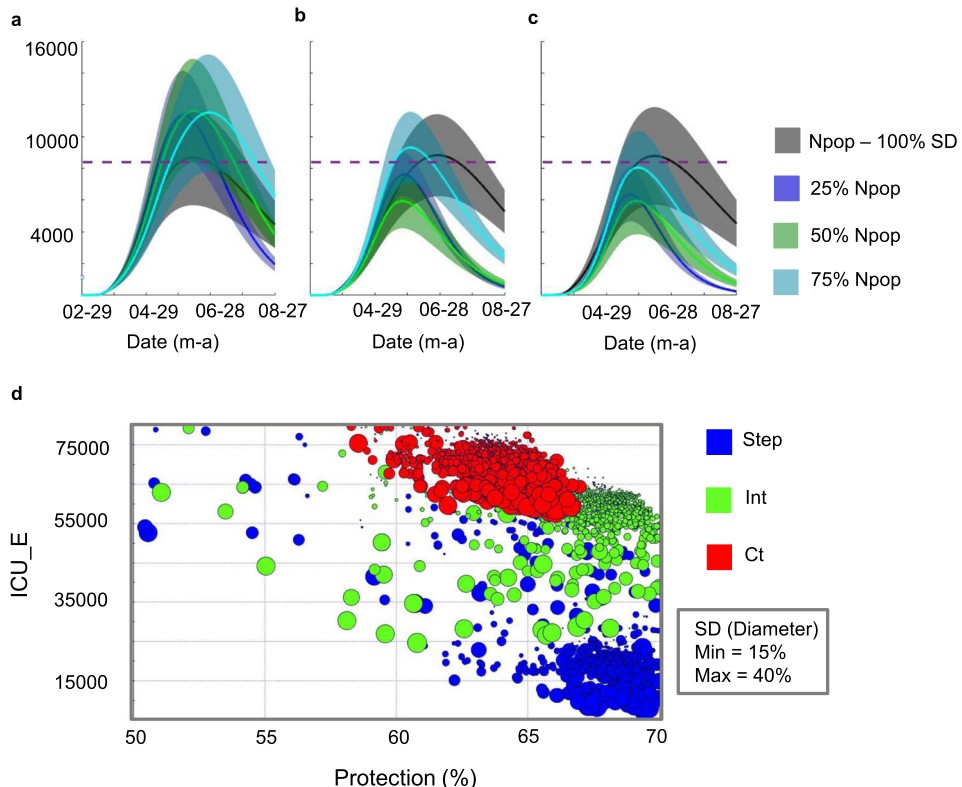

**Fig. 8 Sensitivity analysis considering the impact of having the pandemic spread contained within localized pockets within the state of São Paulo on the progression of the disease estimated by our model and mitigation strategies optimization results for the worst-case scenario investigated in terms of the number of critical cases per day (ICU-PD).** Panel **a** considers having all cases recorded coming from 25, 50, or 75% of the states' population ($N_{pop}$) and isolation has no direct impact in social distancing (SD) magnitude, as estimated by mobility data (100% SD). panel **b** considers isolation causes a decrease of 25% in the effect of SD on the disease's transmission rate (75% SD). Panel **c** considers isolation causes a decrease of 50% in the impact of SD (50% SD). For **a–c** protection rate ($\alpha$) was kept constant across all scenarios and all other model parameters optimized for each scenario and future values of social distancing and protection were chosen to stay at current levels. For comparison, they also show the scenario with no part of the population isolated ($N_{pop}$ and 100% SD). Shaded regions show confidence intervals of 95%, considering the 2.5 and 97.5% quantiles of the distribution of $n = 300$ uniformly distributed 1% errors or perturbations to the model parameters. **d** Mitigation strategy optimization results for 50% isolation and 100% SD showing the impact of different strategies, in color, and the percentage ratio of the unsusceptible or protected people over the whole population (Protection), x-axis, on the total number of critical cases over the ICU threshold (ICU_E), y-axis, predicted till 25 December 2021. Red color indicates constant (Ct) SD strategy, green color indicates intermittent (Int) strategy, and blue color indicates a stepping (Step) down strategy, SD values are shown from 15 to 40% by the diameters of the circles.

understand better the effect in the model of the magnitude of SD on all other parameters, specially $\alpha$. Thus, we further ran the optimization process 20 times, considering 5% decrements to the magnitude of SD on each run. As expected, decreases in SD magnitude were significantly correlated to increases in Protection ($R^2 = 0.72$; $p < 0.001$; Fig. 7b). By using the corresponding regression line, we estimated that considering a drop up to 50% in the magnitude of SD, or analogously consider a drop in the efficiency of mobility to affect $\beta$ by up to 50% (Eqs. 1 and 3), protection levels would increase by approximately 5%. Thus, we used a range of $+5\%$ in our reported results in the text.

Another potential limitation of our model is to assume that the State of São Paulo is a single, homogeneous mixing system; because, in some parts of the state, at a city or neighborhood level, the pandemic spread might be contained within localized pockets. Thus, to verify the impact of possible localized pockets within the state on the disease's progression estimated by our model, we ran a second sensitivity analysis (Fig. 8). We considered having all daily cases and daily deaths recorded within an isolated percentage of the population (25, 50, 75%). Also, we assumed that a more localized spread could result in localized acquired immunity that may influence SD effects on transmission. Figure 8b, c indicates that dropping the impact of SD on the infection rate by 25–50% and having all cases concentrated in 75% of the state yielded similar results to the ones reported. Additionally, the sensitivity analysis indicates that if disease-free isolated regions include more than 25% of the population (e.g. Fig. 8: 25% $N_{pop}$—blue line and 50% $N_{pop}$—green line), our results and projections may be over-estimations. Finally, our results may have been under-estimations if there are localized pockets within the state, but they do not influence the impact of SD (Fig. 8a). To test the reliability of our future projections' conclusions in terms of protection and mitigation SD strategy, we considered the case with the highest estimated peak of critical cases

and ran the strategy optimization workflow (Fig. 8d). The results indicated the same conclusions as the ones described in the "Results" section.

Another limitation of our study was to assume that, when corrected by differences in population's age, the IFR of the disease is constant across all countries, and the lower fatality rate for Iceland is due to higher testing numbers. This factor's main limitation is to assume the same proportion of people with COVID-19 comorbidities across countries. Nevertheless, such limitation has little effect on our study's main conclusions, as investigating IFR was not our focus and would not influence ICU occupancy.

**Reporting summary.** Further information on research design is available in the Nature Research Reporting Summary linked to this article.

## Data availability
COVID-19 daily cases and deaths' time series for Brazil are available from Worldometers.info, Dover, Delaware, U.S.A. [https://www.worldometers.info/coronavirus/][2]. Daily cases and deaths' time series for the state of São Paulo are available from Fundação Sistema Estadual de Análise de Dados Estatísticos (SEADE) [https://www.seade.gov.br/coronavirus/][32]. Mobility data are available from Apple Inc. [https://www.apple.com/covid19/mobility][11] and from Google LLC [https://www.google.com/covid19/mobility][12].

## Code availability
The codes associated with the current analysis are openly available in [GitHub] with the identifier [https://doi.org/10.5281/zenodo.4263126][48].

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

## Acknowledgements

The authors would like to thank Apple, Google, São Paulo State SEADE, Worldometer, and the journals that have made data and information relative to COVID-19 publicly available. We would also like to thank all the frontline workers risking their lives to help others during the COVID-19 pandemic.

## Author contributions

O.P.N., J.C.R., and R.A.Z. conceived and planned the study; O.P.N., G.J.Z., and J.M.d.S. developed the mathematical model. W.P., R.C.d.M.P., B.d.M.B., and E.O.A. worked on revising previous studies and data gathering. O.P.N. designed the computational framework and performed the computations. J.C.R., A.C.B.B., and D.M.K. wrote the introduction. O.P.N. and J.M.d.S. wrote the "Methods" and "Results" sections. O.P.N., D.M.K., and G.J.Z. wrote the discussion. O.P.N., D.M.K., and Y.W. worked on the revision of the paper. All authors made substantial contributions to the conception and design of the work. All authors have discussed, revised, and approved the contents of the final manuscript. All authors have agreed to be accountable for all aspects of the work in ensuring that questions related to the accuracy or integrity of any part of the work are appropriately investigated and resolved.

## Competing interests

The authors declare no competing interests.
