## [Peer Review File · Nature Communications]

REVIEWER COMMENTS

Reviewer #1 (Remarks to the Author):

This paper presents a mathematical model to deal with the new coronavirus in the State of Sao Paulo.

The mathematics proposed in the paper has nothing new and the interest of the results is regional and, therefore, should be submitted to a local journal.

Reviewer #2 (Remarks to the Author):

This paper deals with an important problem which is the epidemiological of COVID-19 in the very populous state of Sao Paulo in Brazil. The authors use a simple mathematical model fitted to the local epidemic, where they assume that the epidemic itself is wholly self-contained (i.e. no interactions with other Brazilian states or other countries). Should the paper be published in this journal I would recommend some substantial changes.

i) The presentation of the data themselves is minimal, showing just the available time series. The authors assume that the state can be treated as a single homogeneously mixing system. It is possible that this is the only data available to them. It would be useful however, to be able to know if their fundamental conclusions are likely to be influenced by this assumption. At last I looked, the number of cases in Sao Paulo had reached 1 in 182. If one considers that it is likely that there are many hidden cases, then the proportion of the population affected may be quite high. Given the importance of the conclusions it would be important to

reasonable to In order to be credible, the authors should at least show that their assumption of a single, homogeneously mixing system is reasonable. This includes showing that there is limited spatial variation in the epidemic progression, and showing evidence that movements between Sao Paulo and other states is not likely to be important.

The former is most important because in their simulations, the number of infected individuals becomes very large, sufficiently so, so that the proportion of people who are likely to be immune in a localised area has a strong influence on the dynamics of infection (an obvious 'false' example would be if all the cases occurred in a single small city of 100,000 individuals, but the rest of the state was disease free, the influence of the cases of COVID-19 would be very different from if the cases were spread all over the state. A geographical representation of case distribution would help to reassure that spatial distribution was not important.

Should the authors show that these factors are unlikely to influence the model outcomes, then the form of the model itself appears reasonable on the whole (but see below) and the intervention strategies considered are both sensible and appropriate.

There is however one critical flaw in the modelling approach, which is the method used to account for social distancing. In the model, the level of social distancing directly modifies the transmission rate. To assess the impact of social distancing, the authors assume its effect is proportional to the changes in mobility presented in the mobility data available from google and Apple. However these figures as provided by google and apple only provide the volume of data of people moving a substantial difference from their home location. It is not a direct proxy for contact rates. The effect of mobility reduction is to (at least in part) shifts potentially infectious contacts to become more local, and more centred around households. For at least the short term, this is likely to have a compensating effect of increasing contact with a smaller number of individuals.

Thus the relationship between the observed mobility and transmission is complicated. This can be seen by comparing the google mobility data to estimates of the reproduction number in the UK (for

example - these are the estimates I know best). While the mobility data shows a very steep drop-off and then a slow, effectively linear rise after lockdown measures are in place, estimates of R (and therefore the influence on transmission) behave very differently, with a drop-off only considerably after the restrictions are in place, and no matching rising in R as mobility comes close to 'normal' again. Because of this poor relationship between R and mobility, this makes projections on the basis of the observed mobility changes problematic.

I accept however, that in the absence of more detailed data, assessing this could be problematic and there is some considerable value in an analysis of what information we have. However, at the very least, the authors should consider as a sensitivity analysis the extreme case where social distancing has no impact on transmission but only 'localises' the epidemic. This therefore is related to the homogeneous mixing assumption as noted above. The effect could be crudely approximated for by assuming that SD breaks Sao Paulo down into smaller independent units and assessing the impact on the analysis of increased levels of local immunity plus the protection effect (α). This would provide a contrasting scenario to the ones the authors present to determine if the combinations of SD and α they propose to prevent a second wave would do so.

Some further specific points follow here:

line 72 - the value of R_0 does not indicate that Brazil does not have the epidemic under control as it explicitly is a value in the absence of control. The authors may mean that it is the value of ' $R(t)$ '? Also the link provided in the reference is to a news item, not to an actual analysis.

line 99 as the authors point out the epidemic in Brazil is very heterogenous, with some states suffering much worse than others. There is no guarantee that similar problems exist at the within state level - therefore having recognised the problem at the national level, it is not clear why they do not address it at the state level, at the very least by sensitivity analysis (see general comment above).

line 107 onwards. the presumed trade-off between social distancing and protection is likely to be flawed for the reasons above - the presumed dependence of SD on mobility puts a lot of emphasis on these changes in mobility when in fact they may not have had a substantial impact at all.

line 110. Because the only levers in the model to change transmission are via SD via protection or via buildup in immunity, its not surprising that in a homogeneously mixing population, there isn't much chance of stopping a second wave without substantial across the state measures.

line 201 - the point about poor testing and tracing clearly applies to the first wave - however it might be useful to note that, should sufficient reduction be achieved, even the relatively low volume and low intensity test and trace in Brazil may have a substantial impact.

line 277 - the model was fitted using accumulated deaths and cases - however this means that the later points disproportionately influences the fit. By eye it doesn't look like this has result in obvious biases but it is very difficult to tell, and at the very least a more rigorous examination of the fit is merited. The model incidence should be plotted as should the residuals over time. Extensive systematic error may suggest a different result should the model be fitted to incidence instead.

line 279. The use of Icelandic figures to fix this value seems surprising, given what the authors have said about the problems of using northern country data for Brazil (see line 86).

line 292. it is possible I have misunderstood what is meant here, but typically I would take the 95% CI's to refer to variation across stochastic simulations. Here a deterministic ODE model is used and I think the variation being considered here is in the parameters - is that correct? Whether it is or not I would fine a better explanation here helpful.

Supplementary information. I have tried to read the report on the optimisation framework several times but have failed to properly understand it. This may be simply because it is not my area of research and to an expert it may be transparent. If a reviewer expert in this field has not seen the paper I would recommend that this be done as I'm not able to assess this aspect of the paper.

Reviewer #3 (Remarks to the Author):

In the proposed paper, the authors use a compartmental model to characterize the spread of SARS-CoV-2 in both Brazil and Sao- Paulo. Given the high attack rate in Brazil, there is strong public health need to understand the epidemic in the country.

The authors present fits to cumulative cases and deaths. As cumulative counts can only increase, this represents a low bar for showing a good fit. Instead the authors should present the fit to daily/weekly data - this also is a clearer way of presenting the course of the epidemic.

It would be useful to set out what the mobility data from Apple/Google is showing - i.e., how much change has there been in mobility over the course of the epidemic.

R_0 is generally only the reproductive number at the start of the epidemic. The authors refer to R_0 being less than 1 in China - at this stage, the virus had circulated for a while - it would be better to refer to R .

It would be useful to set out the underlying Infection Fatality Ratio (IFR) inferred by the model. It should also be made clear whether or not it is assumed to change by age or whether it represents an overall average.

It would be useful to set out the proportion of the population infected over time - and whether the reduction at any moment is due to the impact of immunity or only to the interventions.

It was unclear how 95% confidence intervals were calculated - in line 293 the authors refer to monte Carlo sampling but then refer to 2% error. 95% would normally come from the 2.5% and 97.5% quantiles of the distribution.

Line 292: Remove repeated sentence

Please find below point-by-point responses to the reviewers' comments.

Reply to REVIEWER COMMENTS:

Reviewer #1 (Remarks to the Author):

This paper presents a mathematical model to deal with the new coronavirus in the State of Sao Paulo. The mathematics proposed in the paper has nothing new and the interest of the results is regional and, therefore, should be submitted to a local journal.

Thank you for reviewing our study. We acknowledge the critic, but respectfully disagree. Given the high incidence of COVID-19 infections and deaths in Brazil in general and São Paulo specifically, understanding the effectiveness of intervention strategies on transmission dynamics is of major global health importance. In addition, the results may have universal application. While our analysis focused on São Paulo it is important to note that our model can be used to model other countries, regions, states, cities, and/or organizations. We, however, agree that the importance of our work to global health was not made clear in the previous version of the manuscript. Therefore, we addressed this issue in the current revision. (for examples please see Lines:45-48; 92-94; 256-260 of our revised study)

Although we used a standard epidemiological model as basis for our model, there are several original steps in our proposed method that are new, especially considering them combined. First, we used real mobility data as a time series input to the model; we are not aware of other epidemiological studies that have done so. However, although original, this idea has one drawback which is using people cell-phone mobility as a proxy for contact rates. We addressed this limitation in the current revised version of our study, which I believe may enhance its originality aspect. Additionally, introducing a protective parameter helped divide two possible strategies on dealing with the pandemic and heightened the focus of the mitigation strategies discussed to be plural and not single like in most studies concerning social distancing intermittent strategies. Finally, the mathematics involved in the optimization process to find the optimal solution within the proposed social distance mitigation strategies is also new.

Reviewer #2 (Remarks to the Author):

This paper deals with an important problem which is the epidemiological of COVID-19 in the very populous state of Sao Paulo in Brazil. The authors use a simple mathematical model fitted to the local epidemic, where they assume that the epidemic itself is wholly self-contained (i.e. no interactions with other Brazilian states or other countries). Should the paper be published in this journal I would recommend some substantial changes.

Thank you very much for thorough revision of our study. We worked hard on revising our study and considered every comment that you have given us. We believe your suggestion greatly improved our manuscript.

i) The presentation of the data themselves is minimal, showing just the available time

series. the authors assume that the state can be treated as a single homogeneously mixing system. It is possible that this is the only data available to them.

Thank you very much for the comment. Our model does assume that the State of São Paulo is a single, homogeneously mixing system. However, we added data available for the city of São Paulo and the Inner State, Lines 272 - Figure 6a,b. We used that data to verify the assumption that São Paulo State is a single, homogeneously mixing system. We analyzed how the COVID-19 daily cases and daily deaths in the state as a whole correlated to the data of the city of São Paulo and the rest of the state ($R^2=0.99$, $p<0.001$ for all correlations; Figure 7a,b show the time series). This information has been added to the manuscript.

At last I looked, the number of cases in Sao Paolo had reached 1 in 182. If one considers that it is likely that there are many hidden cases, then the proportion of the population affected may be quite high.

Thank you very much for the comment. We included the attack rate found by our model for Brazil and the state of São Paulo. Please see lines: 88-89. To account for the hidden cases, we considered a high correction factor for the number of cases (please see lines 328-331 of the revised version).

Given the importance of the conclusions it would be important to ... show that their assumption of a single, homogeneously mixing system is reasonable. This includes showing that there is limited spatial variation in the epidemic progression, and showing evidence that movements between Sao Paolo and other states is not likely to be important. A geographical representation of case distribution would help to reassure that spatial distribution was not important.

Thank you for the comment. In addition to the correlation analysis addressed in the first comment to verify the assumption that São Paulo State is a single, homogeneously mixing system, we also use mobility data to demonstrate consistency across the state (please see lines 273-279 Figure 6c.) In addition, we verify our assumption that São Paulo is a single, homogeneously mixing system in the supplementary information. More specifically, we provided information on the population density of the state and showed that São Paulo was relatively isolated from other countries and states within Brazil during the pandemic period using data from state bus stations and airports demonstrating that movement of people for outside the state drastically decreased due to the pandemic: Please see lines 277-279: Supplementary file 1.

There is however one critical flaw in the modelling approach, which is the method used to account for social distancing. ... It is not a direct proxy for contact rates. ... Thus the relationship between the observed mobility and transmission is complicated. This can be seen by comparing the google mobility data to estimates of the reproduction number in the UK (for example - these are the estimates I know best). While the mobility data shows a very steep drop-off and then a slow, effectively linear rise after lockdown measures are in place, estimates of R (and therefore the influence on transmission) behave very differently, with a drop-off only considerably after the restrictions are in place, and no matching rising in R as mobility comes close to 'normal' again. I accept however, that in the absence of more detailed data, assessing this could be problematic and there is some considerable value in an analysis of what information we have.

However, at the very least, the authors should consider as a sensitivity analysis ... This would provide a contrasting scenario to the ones the authors present to determine if the combinations of SD and α they propose to prevent a second wave would do so.

Thank you very much the comment. I believe this commentary had a huge impact in our article; by discussing this aspect and including a sensitivity analyses we greatly strengthened our conclusions. We acknowledge this limitation in our new section “*Model Limitations*” Lines 354-390. To understand the impact of this limitation in our conclusions, first we showed how in our model the magnitude value of social distancing does not have constant effect on our effective reproduction number through time (Lines 357-373 Figure 8a). This aspect is important as it helps modeling the UK example given. In practical terms, the first change in mobility, in the beginning of the pandemic, happened when people were not yet engaged in protective behavior; however, re-opening happened with people increasing mobility, but with much greater caution and knowledge on how avoid getting infected. Additionally, to estimate the real impact of the magnitude of social distancing used in our study in our conclusions we, as suggested, ran a sensitivity analysis (Lines 373-390; Figure 8b). By assuming that we may have overestimate contact rate in our equations by using mobility by a factor of at most 2, we found that protection levels would rise by approximately 5%.

Some further specific points follow here:

line 72 - the value of R_0 does not indicate that Brazil does not have the epidemic under control as it explicitly is a value in the absence of control. The authors may mean that it is the value of ' $R(t)$ '? Also the link provided in the reference is to a news item, not to an actual analysis.

Thanks for the comment. We change R_0 to R_t and fixed the references. Please see lines 59-64.

line 99 as the authors point out the epidemic in Brazil is very heterogenous, with some states suffering much worse than others. There is no guarantee that similar problems exist at the within state level - therefore having recognised the problem at the national level, it is not clear why they do not address it at the state level, at the very least by sensitivity analysis (see general comment above).

Thank you for the comment. We performed the sensitivity analysis, as suggested, to estimate the real impact of the magnitude of social distancing used in our study in our conclusions. More specifically, we ran a sensitivity analysis to better understand the effect of the magnitude of SD on all other model parameters, but specially α and consequently protection. Thus, we further ran the optimization process 20 times considering 5% decrements to the magnitude of SD on each run (Table 2 Line 618).

line 110. Because the only levers in the model to change transmission are via SD via protection or via buildup in immunity, its not surprising that in a homogeneously mixing population, there isn't much chance of stopping a second wave without substantial across the state measures.

Thanks for the comment. We agree; considering the pressure to re-open we believe our results propose a method for lowering SD to doable levels by emphasizing protection; we realized we had not made this as clear in our earlier version of the we

changed our phrasing and better presented this idea, for example, please see lines 121-125 and 208-214; Figure 5.

line 201 - the point about poor testing and tracing clearly applies to the first wave - however it might be useful to note that, should sufficient reduction be achieved, even the relatively low volume and low intensity test and trace in Brazil may have a substantial impact.

Thanks for the comment. We have noted this information in the manuscript please see Lines 208-211.

line 277 - the model was fitted using accumulated deaths and cases - however this means that the later points disproportionately influences the fit. By eye it doesn't look like this has result in obvious biases but it is very difficult to tell, and at the very least a more rigorous examination of the fit is merited. The model incidence should be plotted as should the residuals over time. Extensive systematic error may suggest a different result should the model be fitted to incidence instead.

Thanks for the comment. We changed our fitting procedures as suggested. Now data was fitted to both daily cases and daily deaths time series; also, we included a figure showing the residuals over time; please see lines 327-328 Figure 1.

line 279. The use of Icelandic figures to fix this value seems surprising, given what the authors have said about the problems of using northern country data for Brazil (see line 86).

Thanks for the comment. Given the poor testing rates in many countries, data from Iceland is believed to be a more accurate representation of global cases than from other available sources. We added this information to the text and provided a reference to support this belief. In addition, we discussed this as a limitation of our study. Please see lines (335-337 and 385-390).

line 292. it is possible I have misunderstood what is meant here, but typically I would take the 95% CI's to refer to variation across stochastic simulations. Here a deterministic ODE model is used and I think the variation being considered here is in the parameters - is that correct? Whether it is or not I would fine a better explanation here helpful.

Thanks for the comment. We were not clear in our first version, you are right, we meant variations to the parameters, we included the explanation in our revised version. Please see Lines 346-350.

Reviewer #3 (Remarks to the Author):

In the proposed paper, the authors use a compartmental model to characterize the spread of SARS-CoV-2 in both Brazil and Sao- Paulo. Given the high attack rate in Brazil, there is strong public health need to understand the epidemic in the country.

Thank you very much for reviewing our manuscript. We worked hard on revising our study and considered your comments greatly.

The authors present fits to cumulative cases and deaths. As cumulative counts can only increase, this represents a low bar for showing a good fit. Instead the authors should

present the fit to daily/weekly data - this also is a clearer way of presenting the course of the epidemic.

Thanks for the comment. We changed our fitting procedures as suggested. Now data was fitted to both daily cases and daily deaths time series; please see lines 327-328 Figure 1.

It would be useful to set out what the mobility data from Apple/Google is showing – i.e., how much change has there been in mobility over the course of the epidemic.

Thanks for the comment. As suggested, we included a new figure to show the mobility data as reported by Google and Apple and our treated data used in the model. Please see Line 316-317. Figure 7.

R0 is generally only the reproductive number at the start of the epidemic. The authors refer to R0 being less than 1 in China – at this stage, the virus had circulated for a while – it would be better to refer to R.

Thanks for the comment. We change R0 to Rt. Please see lines 59-64.

It would be useful to set out the underlying Infection Fatality Ratio (IFR) inferred by the model. It should also be made clear whether or not it is assumed to change by age or whether it represents an overall average.

Thank you for the comment. We clarify that now on Line 337.

It would be useful to set out the proportion of the population infected over time – and whether the reduction at any moment is due to the impact of immunity or only to the interventions.

Thank you for the comment. We included the attack rate for São Paulo and Brazil now in the day of the analyses (Please see lines 88-89). Considering less than 4% of the population have been affected till the date of analysis, at this point reduction in daily virus reproduction number has been caused by interventions. We have included a discussion on how both social distancing and protection affects Rt. Please see Lines 355-370 Figure 8a.

It was unclear how 95% confidence intervals were calculated – in line 293 the authors refer to monte Carlo sampling but then refer to 2% error. 95% would normally come from the 2.5% and 97.5% quantiles of the distribution.

Thanks for the comment. We were not clear in our first version, you are right, we meant variations to the parameters, we included the explanation in our revised version. Please see Lines 346-350.

Line 292: Remove repeated sentence

Thanks for the comments, sentence was removed.

REVIEWER COMMENTS

Reviewer #2 (Remarks to the Author):

I thank the authors for their extensive revisions and find most of their response to be useful and greatly improving the manuscript. The revision of the social distancing aspects are very helpful and are particularly improved.

However, in my view the authors have also misunderstood the issues that homogeneous mixing cause. They have shown that the epidemics in the city of Sao Paulo the inner state and more broadly appear to be closely correlated. It is not surprising that they are broadly correlated of course because, in the early seeding phase, there was likely considerable movement between these three areas and therefore they would initially start the same way. However (as is true in many other places) there are also strong reporting biases across dates and thus some of the correlation (e.g. the 'spikes') may in part be due to correlation of these reporting biases (e.g. there is often a weekend bias, which results in apparent spikes and synchronised correlations on Mondays - this is one, but not the only possible correlation caused by reporting). Second, they seem to be making two arguments that are about different things. By showing the correlation they claim that, even after restrictions were in place, Sao Paulo was experiencing homogeneous mixing, rather than being simply correlated due to early synchronisation. On the other hand, they suggest that the available mobility data (showing connections between Sao Paulo and other states) suggest that Sao Paulo was not influenced by other states. It seems to me that they can't have it both ways. The reduction in mobility presumably occurred at scales smaller than Sao Paulo as well (as suggested by the generalised mobility data). Thus if the argument is that Sao Paulo is isolated at the state level because of the mobility reduction it would also suggest that for consistency, they have to say that mobility at some scale below the state also affected transmission - thereby resulting in a violation of their homogeneous mixing assumption. What we need to know is where more localised spread would have resulted in more local increases in acquired immunity that would have likely had a substantial influence on the progression of disease, thereby changing the parameter values that are consistent with the epidemic, and influence the impact of social distancing. Effectively what they need to identify is the scale at which increasing levels of local immunity has an important influence on the course of the infection. To give an extreme example (not realistic) the proportion infected in the population was 10% but all that infection was concentrated in a isolation subgroup that contained 10% of the population (i.e. everyone in the subpopulation became infected), then it would be fair to say that any control measures were effectively useless, whereas if that 10% were spread over a homogeneously mixing population, you might say that control measures were a success. At the very least, the authors should consider another sensitivity analysis to identify when their conclusions would fail. If they can do that, then in my view this paper should be published.

Reviewer #3 (Remarks to the Author):

I have no further comments

Reply to REVIEWER COMMENTS:

Reviewer #2 (Remarks to the Author):

Thank you very much for a second thorough revision of our study. As for the first revision, we worked hard on revising our study and considered the comments that you have given us. We believe your suggestion greatly improved our manuscript.

I thank the authors for their extensive revisions and find most of their response to be useful and greatly improving the manuscript. The revision of the social distancing aspects are very helpful and are particularly improved.

Thanks for acknowledging our revision, we believe your suggestion greatly improved our work.

However (as is true in many other places) there are also strong reporting biases across dates and thus some of the correlation (e.g. the 'spikes') may in part be due to correlation of these reporting biases (e.g. there is often a weekend bias, which results in apparent spikes and synchronised correlations on Mondays - this is one, but not the only possible correlation caused by reporting).

Thanks for the comment. We acknowledge this limitation in our argument in the revised version of the text (lines 273-273). In addition, to strengthen our assumption of a limited spatial variation in the pandemic progression we included a new figure comparing the COVID-19 cases' density map on April 30, May 15, and May 30, 2020 (Figure 7). Based on your comment, we also excluded figure 6c, the figure to not add additional information relevant to the argument at hand and mobility data can be seen in Figure 8.

On the other hand, they suggest that the available mobility data (showing connections between Sao Paolo and other states) suggest that Sao Paolo was not influenced by other states. it seems to me that they can't have it both ways.

The reduction in mobility presumably occurred at scales smaller than Sao Paolo as well (as suggested by the generalised mobility data). Thus if the argument is that Sao Paolo is isolated at the state level because of the mobility reduction it would also suggest that for consistency, they have to say that mobility at some scale below the state also affected transmission - thereby resulting in a violation of there homogeneous mixing assumption. ...

Thanks for the comment. We agree. We include a new paragraph in our model limitations section to address this issue. Please see lines 4011-419 of our newest version. In addition, we ran a sensitivity analysis as you suggested (see below) to verify the impact of possible localized pockets within the state on the progression of the disease estimated by our model.

At the very least, the authors should consider another sensitivity analysis to identify when their conclusions would fail. If they can do that, then in my view this paper should be published.

Thanks very much for the suggestion. We include a new sensitivity analysis in our model limitations section. Please see Lines 420-454 and Figures 10 and 11.

REVIEWERS' COMMENTS

Reviewer #2 (Remarks to the Author):

I thank the authors for their revisions and have no further comments other than to suggest that the additional exploration of homogeneity in Sao Paolo, while what I wanted to see, does make the methodology section rather long and is a bit of a digression. If it could be reduced in the section to a brief statement and the details provided in a separate supplement, it might improve readability. But that is for the authors and editors to decide.